# Proteasomes of Autophagy-Deficient Cells Exhibit Alterations in Regulatory Proteins and a Marked Reduction in Activity

**DOI:** 10.3390/cells12111514

**Published:** 2023-05-30

**Authors:** Qiuhong Xiong, Rong Feng, Sarah Fischer, Malte Karow, Maria Stumpf, Susanne Meßling, Leonie Nitz, Stefan Müller, Christoph S. Clemen, Ning Song, Ping Li, Changxin Wu, Ludwig Eichinger

**Affiliations:** 1Shanxi Provincial Key Laboratory of Medical Molecular Cell Biology, Key Laboratory of Chemical Biology and Molecular Engineering of Ministry of Education, Institutes of Biomedical Sciences, Shanxi University, No. 92 Wucheng Road, Taiyuan 030006, China; 2Center for Biochemistry, Medical Faculty, University of Cologne, Joseph-Stelzmann-Str. 52, 50931 Cologne, Germany; 3CECAD Proteomics Facility, Center for Molecular Medicine Cologne, Medical Faculty, University of Cologne, Joseph-Stelzmann-Str. 26, 50931 Cologne, Germany; 4Institute of Aerospace Medicine, German Aerospace Center (DLR), 51147 Cologne, Germany; 5Institute of Vegetative Physiology, Medical Faculty, University of Cologne, 50931 Cologne, Germany

**Keywords:** autophagy, ubiquitin proteasome system (UPS), *Dictyostelium*, ATG9, ATG16

## Abstract

Autophagy and the ubiquitin proteasome system are the two major processes for the clearance and recycling of proteins and organelles in eukaryotic cells. Evidence is accumulating that there is extensive crosstalk between the two pathways, but the underlying mechanisms are still unclear. We previously found that autophagy 9 (ATG9) and 16 (ATG16) proteins are crucial for full proteasomal activity in the unicellular amoeba *Dictyostelium discoideum*. In comparison to AX2 wild-type cells, ATG9^−^and ATG16^−^ cells displayed a 60%, and ATG9^−^/16^−^ cells a 90%, decrease in proteasomal activity. Mutant cells also showed a significant increase in poly-ubiquitinated proteins and contained large ubiquitin-positive protein aggregates. Here, we focus on possible reasons for these results. Reanalysis of published tandem mass tag-based quantitative proteomic results of AX2, ATG9^−^, ATG16^−^, and ATG9^−^/16^−^ cells revealed no change in the abundance of proteasomal subunits. To identify possible differences in proteasome-associated proteins, we generated AX2 wild-type and ATG16^−^ cells expressing the 20S proteasomal subunit PSMA4 as GFP-tagged fusion protein, and performed co-immunoprecipitation experiments followed by mass spectrometric analysis. The results revealed no significant differences in the abundance of proteasomes between the two strains. However, we found enrichment as well as depletion of proteasomal regulators and differences in the ubiquitination of associated proteins for ATG16^−^, as compared to AX2 cells. Recently, proteaphagy has been described as a means to replace non-functional proteasomes. We propose that autophagy-deficient *D. discoideum* mutants suffer from inefficient proteaphagy, which results in the accumulation of modified, less-active, and also of inactive, proteasomes. As a consequence, these cells exhibit a dramatic decrease in proteasomal activity and deranged protein homeostasis.

## 1. Introduction

Cellular homeostasis is maintained by a precisely regulated balance between synthesis and degradation of cellular components. Autophagy and the ubiquitin proteasome system (UPS) are the two major routes for protein and organelle clearance in eukaryotic cells. Short-lived, abnormal, or damaged proteins are in general degraded by the UPS. In contrast, most of the long-lived proteins, protein aggregates and cellular organelles, as well as invading microorganisms are cleared by macroautophagy (hereafter autophagy) [1,2]. Thus, the two degradative pathways are of utmost importance for the recycling of cellular material and there is increasing evidence for extensive crosstalk between them [3,4,5,6,7]. For example, the core autophagy proteins ATG8 (LC3 in mammals), ATG9, ATG12 and ATG14 can be degraded by the 20S proteasome [8,9,10,11]. Furthermore, treatment of ATG5 knock-out mouse embryonic fibroblasts (MEFs) with proteasome inhibitors resulted in increased protein levels of ATG16, suggesting that ATG16 is also a client of the UPS [12]. Vice versa, proteasomal subunits were found to be degraded by lysosomes [13]. In 2015, Marshall et al. reported that in *Arabidopsis thaliana* whole proteasomes were degraded by a newly described form of selective autophagy, termed proteaphagy [14]. In this pathway, inactive proteasomes were ubiquitinated and targeted for autophagic degradation by RPN10, which binds both ATG8 and the ubiquitinated proteasome [14]. They went on to show that in *Saccharomyces cerevisiae* the autophagic turnover of inactive 26S proteasomes is directed by the ubiquitin receptor Cue5 and the heat shock protein Hsp42, while in HeLa cells the autophagy adaptor protein p62/SQSTM1 delivers the ubiquitinated proteasome to the autophagy machinery through binding to LC3 [15,16]. We found in the social amoeba *Dictyostelium discoideum* that ATG16 mediates the autophagic degradation of proteasomal subunits PSMD1 and PSMD2 through a direct interaction [17,18]. Thus, while proteaphagy appears to be a general mechanism for the degradation of inactive proteasomes, adaptors for performing the task appear to vary from organism to organism [19].

Autophagy and the UPS also functionally interact to maintain cellular homeostasis. It is well established that there is compensatory activation of autophagy upon inhibition of the UPS [3,20,21,22]. In contrast, the effect of autophagy inhibition on the UPS is less clear. Wang et al. reported an upregulation of proteasomal subunits and activity upon pharmacological inhibition of autophagy, and also upon downregulation of autophagy genes by RNAi in colon cancer cells [23]. Komatsu et al. found in ATG7 knock-out mice an accumulation of polyubiquitinated proteins but no obvious alteration in proteasome function [24]. In contrast, it was reported that inhibition of autophagy with chloroquine in neuroblastoma cells resulted in reduced proteasomal activity and in an accumulation of ubiquitinated proteins [25]. Furthermore, mice deficient in the lysosomal enzyme cathepsin D not only had impaired macroautophagy, but also showed reduced proteasomal activity [26]. We found a dramatic decrease in proteasomal activity in several knock-out mutants of core autophagy genes in *D. discoideum* [27,28,29,30,31].

The social amoeba *D. discoideum* is a well-established model organism for the investigation of the autophagic process [32,33]. In this organism, novel conserved autophagy genes have been discovered and the analysis of single, double or triple knock-outs of core autophagy genes revealed informative phenotypes [27,29,30,31,34,35,36,37,38]. Here, we further analyzed the global proteome profiles of wild-type AX2, ATG9^−^, ATG16^−^ and ATG9^−^/16^−^ strains from tandem mass tag (TMT) proteomics data [39]. In addition, we performed co-immunoprecipitation experiments followed by mass spectrometric analysis of AX2 wild-type and ATG16^−^ cells expressing the 20S proteasomal subunit PSMA4 tagged with GFP. The results suggest that proteaphagy is hampered in autophagy-deficient *D. discoideum* mutants. As a consequence, the ratio of less-active or inactive proteasomes increases, causing a dramatic decrease in proteasomal activity and deranged protein homeostasis.

## 2. Materials and Methods

### 2.1. Dictyostelium Strains

*D. discoideum* AX2 was used as wild-type strain. The generation and phenotypes of ATG9^−^, ATG16^−^, ATG9^−^/16^−^, AX2/RFP-PSMD1 and ATG16^−^/RFP-PSMD1 strains are described elsewhere [17,27,37]. The AX2/RFP-PSMD1/PSMA4-GFP and ATG16^−^/RFP-PSMD1/PSMA4-GFP strains were generated by transformation of AX2/RFP-PSMD1 and ATG16^−^/RFP-PSMD1 cells, respectively, with an expression construct encoding GFP fused C-terminally to the PSMA4 subunit of the 20S proteasome. AX2 and mutant cells were grown at 21 °C in HL5 medium (for 1 L: 7.0 g yeast extract, 14.0 g proteose peptone, 13.5 g D-glucose, 0.5 g KH_2_PO_4_, 0.5 g Na_2_HPO_4_) with or without 5 μg/mL Blasticidin S with shaking at 160 rpm in Erlenmeyer flasks or on SM agar plates with *Klebsiella aerogenes* [40,41,42]. The different transgenic strains used in this study are listed in Table 1.

### 2.2. Vector Construction and Transformation

The cDNA encoding full-length PSMA4 (gene ID: DDB_G0280969, http://www.dictybase.org; accessed on 17 September 2018) was amplified by PCR and cloned into the pBsr-C1-GFP vector via the *BamH*I and *Xma*I restriction sites. A linker of nine amino acids with the sequence GGSGGSGGS, which was introduced through the PCR primer, separated PSMA4 from the GFP moiety. The final expression construct was verified by sequencing and introduced into *Dictyostelium* cells by electroporation, and transformants were selected in the presence of 5 μg/mL Blasticidin S. *Dictyostelium* cells expressing PSMA4-GFP were identified by visual inspection under a fluorescence microscope and expression of the fusion protein was verified by Western blotting. The generated strains express PSMA4-GFP, in addition to endogenous PSMA4.

### 2.3. Native Gel Immunoblotting of the Proteasome

A total of 2 × 10^7^ cells were harvested, washed twice in Soerensen buffer and suspended in 200 μL buffer A (50 mM Tris-HCl, pH 7.4, 5 mM MgCl_2_, 10% glycerol (*v*/*v*), 5 mM ATP, 1 mM DTT). Cells were lysed by sonication for 5 s (30% amplitude) and centrifuged at 15,000× *g* for 15 min at 4 °C. The supernatant was normalized for protein concentration using a Bradford protein concentration kit (AR0145, Boster, China). A total of 60 μg of proteins was loaded onto a 3.5% native gel, as described in [44], and separated for 120 min at 120 V. Subsequently, proteins were transferred to a nitrocellulose membrane at 300 mA for 3 h in transfer buffer [45]. Western blot analysis was performed with the anti PSMD2 (1:2000) polyclonal antibody [17]. Three independent experiments were performed for quantification of the proteasome amount in AX2 and ATG16^−^ cells. Normalization was based on actin.

### 2.4. SDS PAGE and Western Blotting

SDS-PAGE and Western blotting were essentially performed as described in [46,47]. *Dictyostelium* total cell lysates were prepared and the proteins of 2 × 10^5^ cells were separated per lane by SDS gel electrophoresis, transferred to a nitrocellulose membrane, blocked with 5% non-fat milk in TBST buffer (10 mM Tris-HCl, pH 8.0, 150 mM NaCl, 0,1% Tween-20) for 60 min at room temperature, then incubated with the primary antibodies. The primary antibodies used were anti-GFP (1:20, K3-184-2), anti-PSMA7 (1:100, 171-337-2), anti-PSMA4 (1:100, 159-183-10), anti-PSMD1 (1:2000), anti-PSMD2 (1:2000), anti-ubiquitin (P4D1 mAb, 1:1000, Cell Signaling Technology, Frankfurt, Germany) and anti-Actin (1:15, Act1-7) [17,48,49,50]. Secondary antibodies used were anti-rabbit and anti-mouse IgG conjugated with horseradish peroxidase (HRP) at a 1:10,000 dilution (Sigma-Aldrich Corp., St. Louis, MO, USA). Detection was performed by chemiluminescence using the SuperSignal West Pico PLUS chemiluminescent substrate (Thermo Scientific Inc., Waltham, MA, USA) in conjunction with the Intas ECL Chemostar documentation system.

### 2.5. Co-Immunoprecipitation (IP) Assays

Co-immunoprecipitation experiments for the identification of differences of the proteasomes and/or proteasome-associated proteins were performed with log phase AX2/RFP-PSMD1 (negative control 1), AX2/RFP-PSMD1/PSMA4-GFP, ATG16^−^/RFP-PSMD1 (negative control 2), and ATG16^−^/RFP-PSMD1/PSMA4-GFP cells (2–4 × 10^6^ cells/mL) cells. A total of 1 × 10^8^ cells were harvested (500× *g*, 5 min), washed twice with Soerensen phosphate buffer (14.6 mM KH_2_PO_4_, 2.0 mM KH_2_PO_4_, pH 6.0) and the cell pellet was shock frozen with liquid nitrogen. Pellets were resuspended in 1 mL lysis buffer (20 mM Tris/HCl pH 7.5, 100 mM NaCl, 1 mM DTT, 20 mM MgCl_2_, 5% glycerol, 1 mM benzamidine, 10 µg/mL aprotinin/leupeptin, 1:50 proteinase inhibitor cocktail (Roche), 1:100 PEFA block) followed by centrifugation at 20,000× *g* for 10 min. Under these conditions the ubiquitin positive aggregates in ATG16^−^ cells are not solubilized. The supernatant, containing soluble proteins, was incubated with GFP-trap beads (Chromotek, Planegg, Germany) for 2 h at 4 °C. Beads were washed four times with wash buffer I (50 mM Tris/HCl pH 7.5, 150 mM NaCl, 1 mM DTT, 0.2% NP-40) and twice with wash buffer II (50 mM Tris/HCl pH 7.5, 150 m NaCl, 1 mM DTT). Bound proteins were either analyzed by SDS-PAGE and silver staining or further processed for mass spectrometry. Co-immunoprecipitations were performed with three independent biological replicates in each condition.

### 2.6. Mass Spectrometry

Mass spectrometry was carried out at the CECAD/CMMC Proteomics Facility (University of Cologne). Samples were prepared by the in-solution digestion of proteins and StageTip purification of peptides, according to the protocol of the facility (http://proteomics.cecad-labs.uni-koeln.de/Protocols.955.0.html). The samples were analyzed using an EASY nLC 1000 HPLC (Thermo Scientific, Waltham, MA, USA) coupled to a Q-Exactive Plus (Thermo Scientific, Waltham, MA, USA) mass spectrometer. Peptides were loaded with solvent A (0.1% formic acid in water) onto an in-house-packed analytical column (50 cm × 75 μm I.D., packed with 2.7 μm C18 Poroshell beads, Agilent, Santa Clara, CA, USA). They were separated at a constant flow rate of 250 nL/min, using a 50 min gradient followed by a 10 min wash with 95% Solvent B (0.1% formic acid in 80% acetonitrile). The mass spectrometer was operated in data-dependent acquisition mode, where the Orbitrap acquired full MS scans (300–1750 *m*/*z*) at a resolution of 70,000 with an automated gain control (AGC, Tokyo, Japan) target of 3 × 10^6^ ions collected with 20 ms. Precursors were dynamically excluded for 20 s. The ten most intense peaks were subjected to HCD fragmentation. All mass spectrometric raw data were processed with Maxquant (version 1.5.3.8) and its implemented Andromeda search engine [19].

MS2 spectra were searched against the *D. discoideum* proteome database (UP000002195, 12,746 entries, downloaded at 23 July 2018) and a list of common contaminants. False discovery rates on protein and PSM level were estimated by the target–decoy approach to 1% (Protein FDR) and 1% (PSM FDR), respectively. The minimal peptide length was set to 7 amino acids and carbamidomethylation at cysteine residues was considered as a fixed modification. Oxidation (M) and Acetyl (Protein N-term), Acetyl (K), GlyGly (K), Phospho (STY) and ubiquitination were included as variable modifications. The match-between-runs option was enabled within replicate groups. LFQ quantification was enabled using default settings. Search output was loaded into Perseus (version 1.6.5.0). Decoys and potential contaminants were removed and the log2 transformed LFQ intensities were filtered for at least 3 out of 3 values in at least one condition. Remaining missing values were imputed with random values from the lower end of the intensity distribution using Perseus defaults and a two-sample two-tailed Student’s t-test was calculated for pairwise comparisons.

The mass spectrometry proteomics data have been deposited at the ProteomeXchange Consortium (http://www.proteomexchange.org/) via the PRIDE [33] partner repository, with the dataset identifier PXD039713.

### 2.7. Miscellaneous Methods

RNA isolation and cDNA generation were essentially carried out as described in [51]. The following primers were used for amplifying the full-length coding sequence of PSMA4. psma4-FL_BamHI_F CGCGGATCCATGGCAAGAAGATATGATCAAAGAACAAC; psma4_Xma_R CGCCCCGGGGCTGCCTCCGCTGCCCCCGCTGCCTCCATTATCTTCAGTTTCTTGTTCGAGATCAG. For quantification of Western blot bands, densitometry analyses were performed with ImageJ v1.50h. Statistical significance of differences is based on Student’s *t*-test. Error bars indicate standard deviations (SD) of the mean of at least 3 independent experiments (* *p* < 0.05, ** *p* < 0.01).

## 3. Results

### 3.1. Autophagy Mutants Exhibit a Strong Decrease in Proteasomal Activity but No Quantitative Change in Proteasomal Proteins

We previously reported that *D. discoideum* knock-out mutants of the core autophagy proteins ATG5, ATG8a, ATG9, ATG12 and ATG16 exhibit a significant decrease in proteasomal activity. Mutant cells also displayed an increase in global protein ubiquitination and cytoplasmic ubiquitin-positive protein aggregates. In contrast to expectation, quantitative Western blot analyses revealed no change in the amount of the 20S proteasomal subunits A4 or A7 [27,28,29,30,31]. We concluded from these results that the decrease in proteasomal activity was somehow caused by the defect in canonical autophagy and not due to a decrease in the number of proteasomes. However, this conclusion was only based on the quantification of the 20S proteasomal subunits, PSMA4 and PSMA7. To substantiate our finding, we performed TMT (tandem mass tag) quantitative proteomic analysis of AX2 wild-type cells, the single ATG9 and ATG16 knock-out and the ATG9/16 double knock-out strains [39].

Analysis of the abundance of all 26S proteasomal subunits revealed no significant differences between AX2 wild-type and the single ATG9 and ATG16 knock-out cells (Appendix A) or the ATG9/16 double knock-out cells (Figure 1a). We confirmed the result for the double knock-out strain for PSMD1and PSMD2 from the 19S regulatory particle and for PSMA7 from the 20S catalytic particle by Western blotting (Figure 1b). Next, we analyzed proteasome abundance in AX2 wild-type and ATG16^−^ strains by native PAGE and Western blotting. The results suggest a moderate increase of approximately 30% of fully assembled 26S proteasomes in ATG16^−^ cells in comparison to AX2 wild-type cells (Figure 1c,d). We obtained similar results for ATG9^−^ and ATG9^−^/16^−^ cells.

Thus, we were confronted with the situation in which our knock-out strains of core autophagy genes harbored a similar number of proteasomal subunits as the AX2 wild-type strain and even a moderately increased number of fully assembled proteasomes, but exhibited a strongly decreased proteasomal activity, which was, e.g., for the ATG9^−^/16^−^ strain only around 10% as compared to AX2 wild-type cells [27].

### 3.2. Immunoprecipitation of Tagged Proteasomes and Mass Spectrometry Support the Finding of Similar Amounts of Proteasomes in AX2 and ATG16^−^ Cells

To shed light on the underlying cause of these results, we decided to generate strains that express a tagged 20S proteasomal subunit for immune precipitation (IP) experiments followed by mass spectrometry. For this purpose, we transformed the previously generated AX2 and ATG16^−^ cells that express RFP-PSMD1 [17] with a construct for the expression of PSMA4-GFP (Table 1). Western blotting confirmed expression of the GFP-tagged PSMA4 in AX2/RFP-PSMD1 and ATG16^−^/RFP-PSMD1 cells, in addition to endogenous PSMA4 (Figure 2). Expression of RFP-PSMD1 was confirmed previously [17]. We also analyzed lysates of the generated strains for global ubiquitination of proteins with the commercially available P4D1 antibody, which recognizes mono-ubiquitin, poly-ubiquitin and ubiquitinated proteins, and cross-reacts with *Dictyostelium* ubiquitin. Consistent with previous results, we observed a significant increase of poly-ubiquitinated proteins in both ATG16^−^ strains either expressing tagged PSMA4 or not, in comparison to AX2 wild-type cells (Appendix A). This result supports the decrease in proteasomal activity in these strains, and is indicative for the absence of an effect on proteasomal activity through expression of tagged PSMA4.

Next, we performed GFP-trap IP with total cell lysates of the AX2 and ATG16^−^ cells either expressing PSMA4-GFP or not (negative controls). Silver staining showed approximately equal amounts of proteins in the total cell lysates of all four strains and distinct bands in the immune precipitates of AX2 and ATG16^−^ expressing PSMA4-GFP. These bands corresponded in size to the 20S proteasome subunits and to PSMA4-GFP (Figure 3a). Western blotting with anti GFP and anti PSMA4 antibodies confirmed the presence of PSMA4 and PSMA4-GFP in the immune precipitates of the strains expressing tagged PSMA4 (Figure 3b). Silver stains and Western blots suggested approximately similar amounts of proteasomes in wild-type AX2 and ATG16^−^ cells (Figure 3).

To further analyze the precipitated proteasomes and associated proteins in AX2 and ATG16^−^ cells, we performed mass spectrometry of three independent experiments for each strain. The experimental replicates worked very well, as shown by principal component analysis (PCA) and heat map analysis, which revealed that the replicates of each strain were most closely related (Appendix A). Label-free quantification (LFQ) showed a strong enrichment of up to 2000-fold for all 20S subunits, all 20S proteasome assembly factors, the proteasome maturation protein homolog (POMP), and all 19S subunits in the immune precipitates of the AX2 wild-type strain and the ATG16^−^ strain expressing PSMA4-GFP, in comparison to the negative controls (Appendix A). The enrichments for both experiments versus negative controls were for nearly all subunits statistically highly significant (Appendix A). For the few subunits where this was not the case, there was a strongly deviating value in one of the biological replicates (not shown).

Volcano plots of the detected proteins revealed that all subunits of the 20S proteasome (light green circles), the 20S assembly chaperones (green circles), and of the 19S proteasome (light blue and blue circles) were enriched more than three-fold in the AX2 and the ATG16^−^ strains, in comparison to the negative controls (Figure 4a,b). Furthermore, we found that the proteasome activator 28 (PSME3), the proteasome activator complex subunit 4 (PSME4), and the proteasome-associated protein ECM29 homolog (orange circles) were enriched more than three-fold in the AX2 and the ATG16^−^ strains, in comparison to the negative controls (Figure 4a,b, Appendix A). In agreement with our Western blot results, none of the proteasomal subunits and the other proteins mentioned above were significantly enriched in the ATG16^−^ versus AX2 comparison of detected proteins (Figure 4c). We noticed, however, that for most 20S subunits and assembly chaperones the enrichment in the experiment versus negative control was generally a little bit higher for ATG16^−^/RFP-PSMD1/PSMA4-GFP, as compared to AX2/RFP-PSMD1/PSMA4-GFP cells. This is reflected in a 1.23- to 2.06-fold enrichment of these subunits in the ATG16-versus-AX2 comparison. However, for 17 of the 19 subunits these differences were statistically not significant (*p*-value > 0.05) (Figure 4c, Appendix A). We conclude that the strong decrease in proteasomal activity in the ATG16^−^ strain is not due to a decrease in the number of proteasomes in this strain. Based on our results, the ATG16^−^ strain harbors at least similar amounts of fully assembled 26S proteasomes (Figure 1c,d and Figure 4; Appendix A).

Next, we looked for differences in the enrichment of proteins for AX2 and ATG16^−^ cells (Appendix A). In comparison to the negative controls, we noticed for AX2 eight uniquely enriched proteins (Figure 4a, dark yellow circles) and for ATG16^−^ seven proteins (Figure 4b, dark red circles). The latter seven proteins were also enriched in the ATG16^−^-versus-AX2 comparison; however, for three of them the statistical significance was slightly above the threshold *p*-value of 0.05 (marked with a star in Appendix A). The most interesting differentially enriched proteins were the shuttle factor ubiquilin, the proteasome inhibitor PI31 (PSMF1), the ubiquitin system component CUE domain-containing protein, and the ubiquitin carboxyl-terminal hydrolase isozyme L5. The latter three were enriched in ATG16^−^ but not AX2 precipitates (Figure 4). Their differential association with the proteasome could be in part responsible for the reduced proteasomal activity in ATG16^−^ cells (see Section 4).

The comparison of co-precipitated proteins in the ATG16^−^/RFP-PSMD1/PSMA4-GFP versus AX2/RFP-PSMD1/PSMA4-GFP samples revealed forty-four enriched proteins. Of these forty-four proteins, four were also enriched in the comparison of ATG16^−^/RFP-PSMD1/PSMA4-GFP versus negative control, and the other 40 were exclusive to the former comparison (Figure 4c, lilac circles; Appendix A). Their exclusive enrichment in the precipitated samples of ATG16^−^/RFP-PSMD1/PSMA4-GFP versus AX2/RFP-PSMD1/PSMA4-GFP cells is caused by their underrepresentation in the precipitated proteasomes from AX2/RFP-PSMD1/PSMA4-GFP cells. The ID of those proteins, which are at least 8-fold enriched in ATG16^−^ versus AX2, is provided in Figure 4c.

### 3.3. Some Proteasome-Associated Proteins Were Ubiquitinated

Our results so far suggest that the decreased proteasomal activity in autophagy-deficient cells was not due to a reduction in the number of proteasomes. However, differential association of regulatory proteins might at least in part be responsible (Figure 4, Appendix A). As a further possibility for regulation of proteasomal activity, we considered post-translational modifications, and looked for differential ubiquitination of proteins in our immune precipitates of the ATG16^−^/RFP-PSMD1/PSMA4-GFP and AX2/RFP-PSMD1/PSMA4-GFP strains. In total, we found 25 proteins in the immune precipitates which were ubiquitinated either in AX2 or ATG16^−^ cells, or in both (Table 2). Three proteins were only ubiquitinated in the AX2 wild-type, six proteins in the precipitates of both strains, albeit with different intensities, and sixteen only in the ATG16^−^ cells. Of the three proteins which were exclusively ubiquitinated in AX2, two are uncharacterized and the third is the 60S ribosomal protein L40. Of the six proteins which were ubiquitinated in both strains, two are uncharacterized. The other four are the 60S ribosomal protein L24, the ubiquitin-60S ribosomal protein L40, the BolA-like protein, and the proteasome subunit beta type-7. The log2 intensities were for all six proteins higher in ATG16^−^ cells. Of the sixteen proteins which were exclusively ubiquitinated in ATG16^−^ cells, twelve are uncharacterized. The remaining four are the eukaryotic translation initiation factor 4E, the T-complex protein 1 subunit alpha, the bifunctional purine synthesis protein purC/E, and the DDI1 homolog, a ubiquitin-associated (UBA) domain-containing protein (Table 2).

## 4. Discussion

The two major systems for cellular homeostasis are autophagy and the ubiquitin proteasome system (UPS) [52]. For a long time, it was thought that these intracellular pathways for protein and organelle clearance act independently, but increasing evidence strongly suggests that they are interrelated [3,4,5,6,53]. An early finding showed that treatment of rats with the lysosomal inhibitor leupeptin resulted in the accumulation of proteasomes in rat liver lysosomes, suggesting the degradation of proteasomes by autophagy [13]. In colon cancer cells, downregulation of autophagy genes by RNAi and pharmacological inhibition of autophagy resulted in an increase of proteasomal subunits and in proteasomal activity, consistent with a compensatory up-regulation of proteasomal activity [23]. In contrast, inhibition of lysosomal activities by chloroquine in neuroblastoma cells led to an accumulation of ubiquitinated proteins and reduced proteasomal activities [25]. Furthermore, inhibition of autophagy with Bafilomycin A1 or siRNA knock-down of *atg7* or *atg12* resulted in an impaired clearance of UPS clients in HeLa cells, consistent with reduced proteasomal activity [4]. In previous work, we found a significant decrease of proteasomal activity in autophagy-compromised *D. discoideum* strains [27,28,29,30,31,32]. These results could either be explained by a decrease in the abundance of proteasomal subunits or a down-regulation of proteasomal activity. The former possibility is unlikely, as whole genome transcriptional analyses and Western blot analyses of selected proteasomal subunits revealed no change in comparison to wild-type cells [27,28,29,30,31]. Further analysis of ATG9^−^, ATG16^−^, and ATG9^−^/16^−^ cells by TMT quantitative proteomics confirmed this result (Figure 1 and Appendix A) [39].

To further shed light on this conundrum we generated AX2 wild-type and ATG16^−^ cells that express the proteasomal subunit PSMA4 tagged with GFP. In our mass spectrometry results of immune-precipitated proteasomes we detected all 19S and all 20S proteasomal subunits and, in addition, all 20S assembly chaperones and 20S activators. The data provided no explanation for the observed reduced proteasomal activity, as there was no indication of a reduced number of assembled proteasomes in the ATG16^−^ strain. In contrast, comparing the abundance of the proteasomal subunits of AX2 with ATG16^−^ cells points to a possible increase of fully assembled proteasomes in the latter strain. This result was, however, for all subunits except three, statistically not significant (Figure 4; Appendix A). Apart from all proteasomal subunits, we detected a number of proteasome-associated proteins in both strains. Eight proteins were only statistically significantly enriched in AX2-versus-negative control, but not in ATG16^−^ cells (Figure 4a, dark yellow circles; Appendix A upper part). For seven of these proteins it is at present unclear whether or how they might be involved in regulating proteasomal activity. The eighth protein, ubiquilin, is one of the three shuttle factors in *D. discoideum*. Ubiquilin acts by binding proteins tagged with polyubiquitin chains and delivers them to the UPS for degradation [54]. Its underrepresentation on proteasomes of ATG16^−^ cells, although slightly not statistically significant because of one strongly deviating value, could result in reduced feeding of the proteasome in this strain. Seven proteins were statistically significantly enriched in ATG16-versus-negative control and in ATG16-versus-AX2 comparisons (Figure 4b,c, dark red circles; Appendix A, lower part). For four of these proteins, the functional connection to proteasomal activity is at present unclear, while the remaining three proteins might play a role in the reduced proteasomal activity of ATG16^−^ cells. The proteasome inhibitor PI31, also known as PSMF1, was identified more than 30 years ago as an inhibitor of the 20S core particle, and recently its mode of inhibiting the core particle was elucidated by cryoEM [55,56,57]. We noticed a more than 16-fold enrichment of PI31 with proteasomes of ATG16^−^ versus AX2 cells, which could be a major reason for the reduced proteasomal activity in ATG16^−^ cells. Furthermore, the ubiquitin carboxyl-terminal hydrolase isozyme L5, which belongs to the set of deubiquitinating proteases (DUBs), was enriched more than 10-fold with proteasomes of ATG16^−^, in comparison to AX2. The *D. discoideum* protein is highly conserved across species, and shares 64% similarity with its human orthologue, which has a potential role in oncogenesis [58]. DUBs suppress protein degradation by deubiquitination of distinct substrates [59]. The enrichment of the ubiquitin carboxyl-terminal hydrolase with proteasomes of ATG16^−^ versus AX2 cells likely contributes to the reduced proteasomal activity in ATG16^−^ cells. Finally, a CUE domain-containing protein was enriched approximately 14-fold with proteasomes of ATG16^−^ cells. The CUE domain is a ubiquitin-binding domain, and recently, yeast Cue5 was identified as a proteaphagy receptor for inactivated 26S proteasomes [16,60]. Proteaphagy, as a clearing mechanism for inactive 26S proteasomes, was first reported in *Arabidopsis thaliana*, where it is mediated by the proteasomal subunit RPN10 (PSMD4) via ubiquitin and ATG8 [14]. In *S. cerevisiae*, proteaphagy is mediated via the ubiquitin receptor Cue5 and the Hsp42 chaperone, and in mammals, stress-induced proteaphagy was dependent on p62/SQSTM1 and ubiquitin [15,16]. Thus, clearance of inactive proteasomes appears to be a general mechanism, albeit the mode of action seems to vary between organisms. Notably, in our TMT quantitative proteomic approach, we detected an increase of the autophagy receptor p62/SQSTM1 and of ATG8b in ATG9^−^/16^−^ cells [27]. Whether these proteins and/or the CUE domain-containing protein are involved in proteaphagy in *D. discoideum* is the subject of future work. The dramatic enrichment of the CUE domain-containing protein at proteasomes of ATG16^−^ cells is consistent with an enrichment of inactivate 26S proteasomes in ATG16^−^ versus AX2 cells (Figure 4; Appendix A).

In addition, our analysis revealed forty proteins, which were more than three-fold enriched with proteasomes in the comparison of ATG16^−^ versus AX2 cells, but not in the comparison of ATG16^−^ cells versus the negative control (Figure 4c, lilac circles; Appendix A). A closer examination of this result showed that thirty-three of these proteins were underrepresented in ATG16^−^ cells versus NC2 (Appendix A, light orange background), and for the remaining seven the generally moderate enrichment in ATG16^−^ cells versus NC2 was statistically not significant (Appendix A, light blue background). Thus, we consider the enrichment of these 40 proteins as not significant for the observed decreased proteasomal activity in ATG16^−^ cells.

Ubiquitination is crucial for delivering clients to the UPS, and also plays a major role in autophagy. Therefore, we also considered differences in ubiquitination of proteasome-associated proteins as a possibility for inhibition of proteasomal activity in ATG16^−^ cells. We detected 25 ubiquitinated proteins in the immune precipitates of AX2 and/or ATG16^−^ cells. Three proteins were only ubiquitinated in the AX2 wild-type, six proteins in the precipitates of both strains, albeit with different intensities, and sixteen only in the ATG16^−^ cells (Table 2). Unfortunately, most of these proteins, i.e. 16 of the 25, are uncharacterized. For seven of the remaining nine the link to the observed decrease in proteasomal activity in autophagy-deficient strains is at present unclear. However, the differential ubiquitination of two proteins, namely PSMB7 and DDI1, could in part be responsible for the lower proteasomal activity in ATG16^−^ cells. We see a significant enrichment of ubiquitinated PSMB7, which is part of the β-ring of the 20S core particle and harbors the trypsin-like activity, in ATG16^−^ in comparison to AX2 (Table 2, bold). Previously, it was shown that subunits of the 26S proteasome were ubiquitinated upon addition of proteasome inhibitors [61]. We further observed exclusive ubiquitination of the DDI1 homolog, a ubiquitin-associated (UBA) and ubiquitin-like (UBL) domain-containing protein in ATG16^−^ (Table 2, bold). DDI1 has been proposed to serve as shuttling factor that delivers ubiquitinated substrates to the proteasome [62]. Recent evidence suggests that DDI1 family members likely cleave poly-ubiquitinated substrates under conditions where proteasome function is compromised [63]. In the case of this being the real cellular function of DDI1, the association of ubiquitinated DDI1 with proteasomes of ATG16^−^ cells would support the accumulation of activity-compromised proteasomes in this and other autophagy-deficient strains.

In summary, we are left in ATG16^−^ and other autophagy-deficient cells with the situation of unchanged abundance of proteasomal subunits concomitant with a dramatic decrease in activity. Our data support a model suggesting that in these cells proteaphagy is hampered and the exchange of malfunctioning proteasomes is strongly inhibited. As a serious consequence, proteasomes in these strains accumulate post-translational modifications, which either per se cause a decrease in activity or constitute additional binding sites for proteasomal activity inhibiting proteins, as for example PI31 (Figure 5). We propose that the overall decrease in turnover of “aged” and less-active proteasomes results in their relative abundance in autophagy-deficient strains, with the observed fatal consequences for proteasomal activity and cellular homeostasis.

## 5. Conclusions

Autophagy-deficient strains suffer from a strong decrease in proteasomal activity although the amount of proteasomal subunits is unchanged and fully assembled proteasomes might even be slightly increased. We propose that due to defective proteaphagy in these cells the regeneration of malfunctioning proteasomes is strongly inhibited. The overall decrease in the turnover of post-translationally modified, i.e., “aged” and less-active proteasomes, leads to their relative abundance and causes the observed fatal consequences for proteasomal activity and cellular homeostasis in autophagy-deficient strains.

## Figures and Tables

**Figure 1 cells-12-01514-f001:**
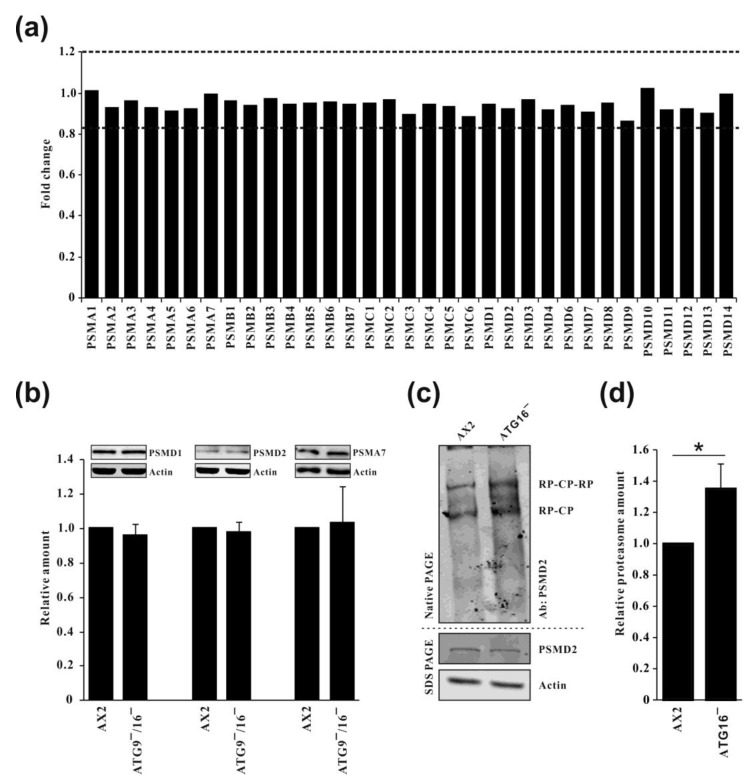
Quantification of all proteasomal proteins and of fully assembled proteasomes. (**a**) Quantification of proteasomal subunits in the ATG9^−^/16^−^ strain in comparison to AX2 based on TMT proteomic analysis. The value of AX2 was set to 1. (**b**) Confirmation of TMT proteomic results for PSMD1, PSMD2, and PSMA7, using Western blots. Mean values and standard deviations of four independent experiments are depicted. (**c**) Native Page and SDS PAGE of cell lysates from AX2 and ATG16^−^ cells, followed by Western blot. **Upper part**, Native PAGE. Fully assembled proteasomes were detected with antibodies against PSMD2. **Lower part**, SDS PAGE. Detection of PSMD2 with a polyclonal antibody and actin with the monoclonal Act1-7 antibody. CP, catalytic particle; RP, regulatory particle. (**d**) Quantification of fully assembled proteasomes in AX2 and ATG16^−^ cells. Three independent experiments were performed. Normalization was based on actin. * *p* ≤ 0.05.

**Figure 2 cells-12-01514-f002:**
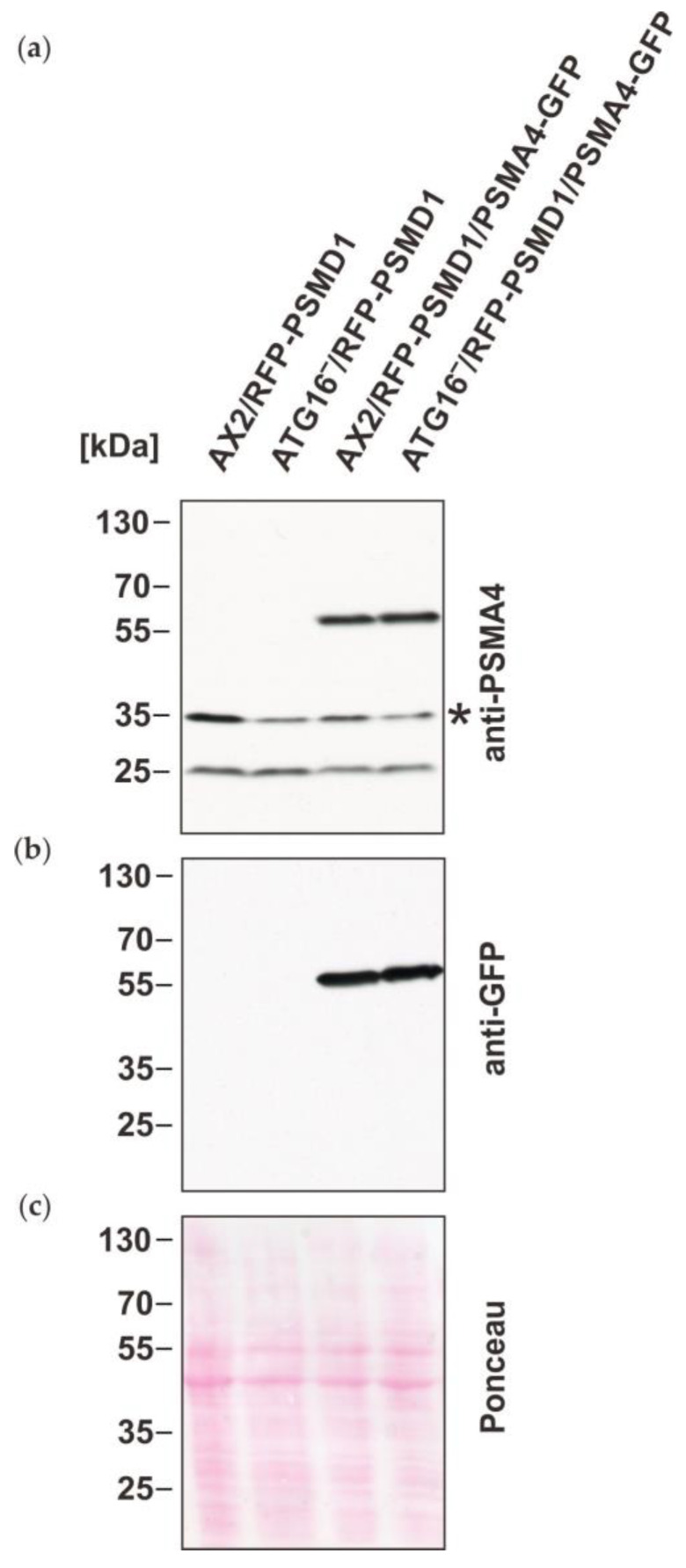
Verification of the expression of PSMA4-GFP in the different mutant strains by immuno-blotting. (**a**) Immunoblotting of total cell lysates of AX2/RFP-PSMD1, ATG16^−^/RFP-PSMD1, AX2/RFP-PSMD1/PSMA4-GFP, and ATG16^−^/RFP-PSMD1/PSMA4-GFP strains. Endogenous PSMA4 and PSMA4-GFP were detected with the PSMA4 mAb [49]; * cross-reaction of the antibody. (**b**) The tagged PSMA4 in the AX2/RFP-PSMD1/PSMA4-GFP and ATG16^−^/RFP-PSMD1/PSMA4-GFP strains was detected with the GFP mAb [48]. (**c**) Ponceau stain of the blotting membrane as a loading control.

**Figure 3 cells-12-01514-f003:**
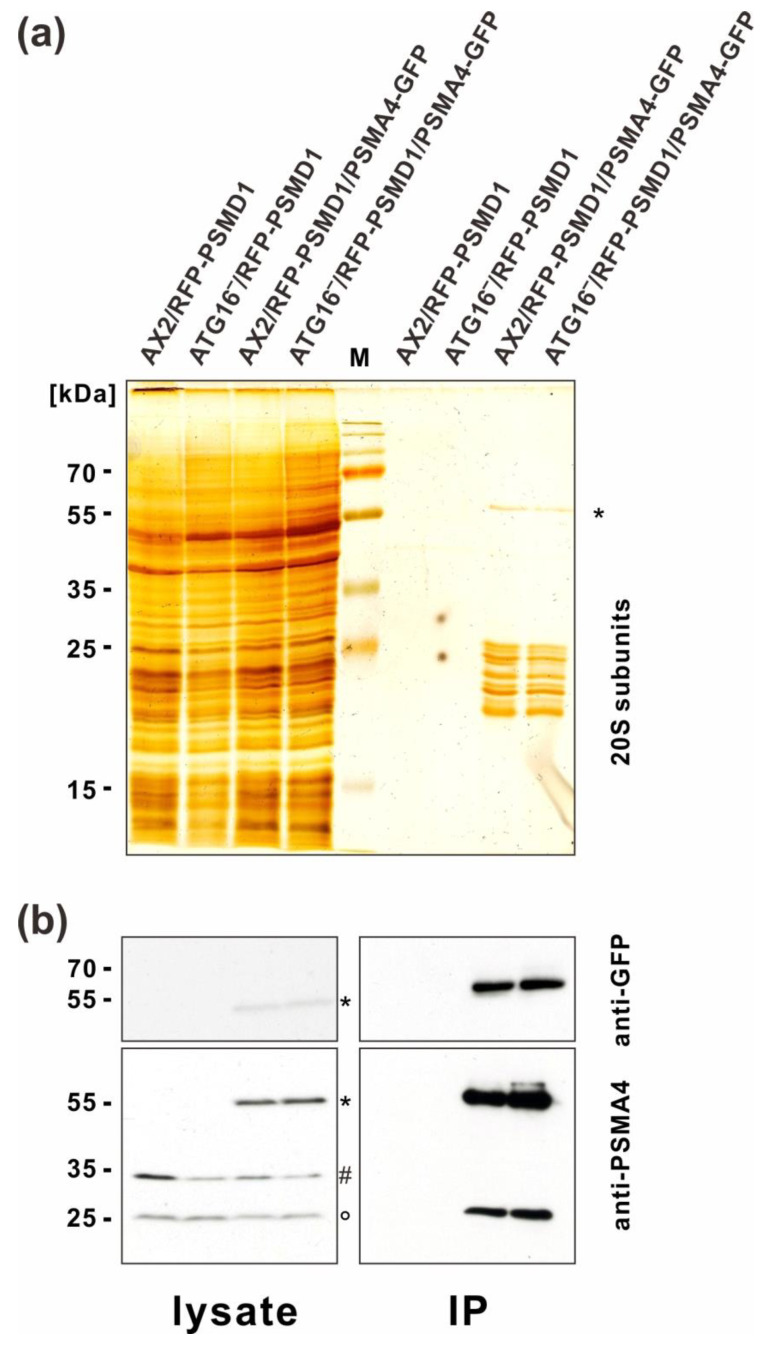
Analysis of GFP-trap immunoprecipitations. (**a**) Silver stain of total cell lysates (**left**) and of immune precipitates (**right**) of AX2/RFP-PSMD1, ATG16^−^/RFP-PSMD1, AX2/RFP-PSMD1/PSMA4-GFP, and ATG16^−^/RFP-PSMD1/PSMA4-GFP strains. The positions of PSMA4-GFP (*) and of 20S proteasomal subunits are indicated. (**b**) Immunostaining of total cell lysates (**left**) and of immune precipitates (**right**) of AX2/RFP-PSMD1, ATG16^−^/RFP-PSMD1, AX2/RFP-PSMD1/PSMA4-GFP, and ATG16^−^/RFP-PSMD1/PSMA4-GFP strains. **Top panel**: anti-GFP immunoblot. **Bottom panel**: anti PSMA4 immunoblot. The positions of PSMA4-GFP (*) and of PSMA4 (°) are indicated. (#): Cross-reaction of the PSMA4 mAb. M: Low molecular weight protein marker.

**Figure 4 cells-12-01514-f004:**
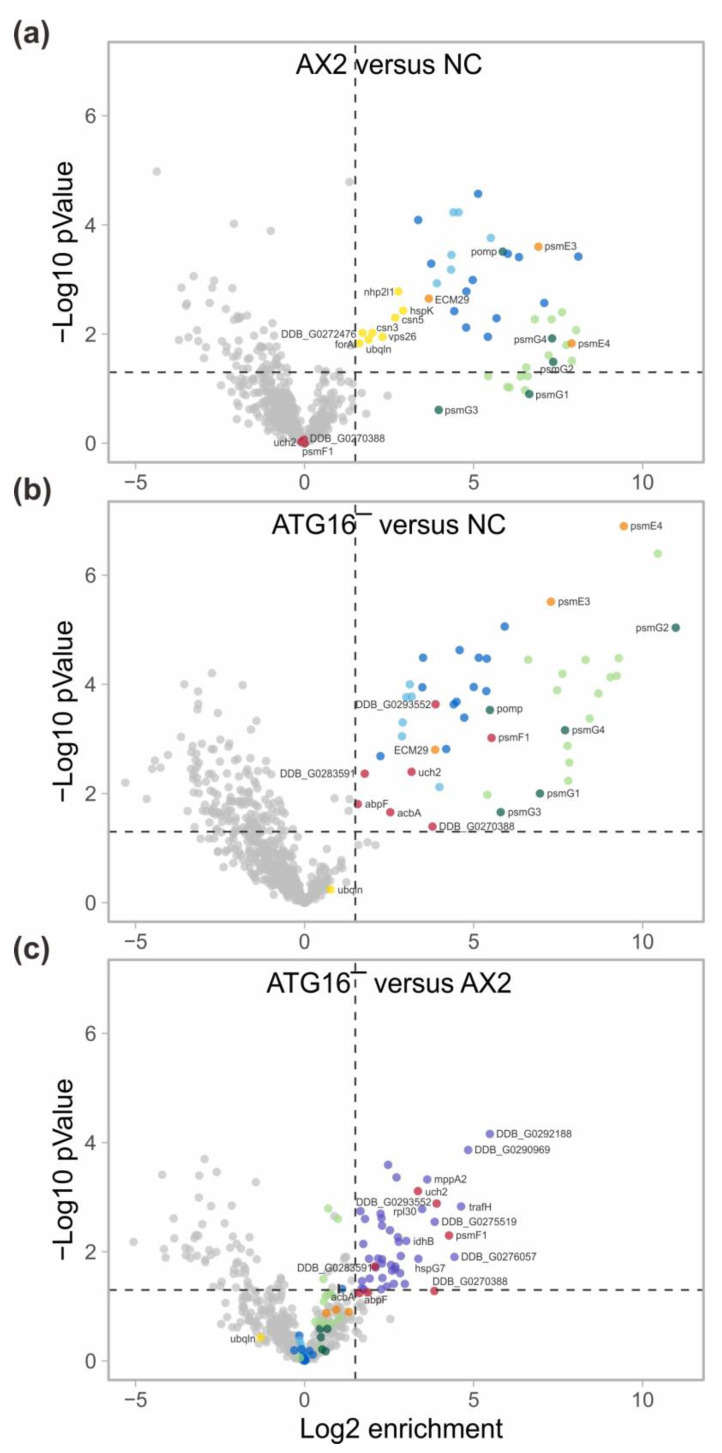
Volcano plots of mass spectrometry results of immune precipitated proteasomes. (**a**) Enriched proteins in AX2/RFP-PSMD1/PSMA4-GFP versus AX2/RFP-PSMD1 (NC) cells. (**b**) Enriched proteins in ATG16^−^/RFP-PSMD1/PSMA4-GFP versus ATG16^−^/RFP-PSMD1 (NC). (**c**) Enriched proteins in ATG16^−^/RFP-PSMD1/PSMA4-GFP versus AX2/RFP-PSMD1/PSMA4-GFP. NC, negative control. 20S subunits are depicted in light green (A- and B-rings) and 20S assembly chaperones and the maturation homolog in dark green. 19S subunits are depicted in light blue (ATPase ring) and blue (lid and regulatory proteins). Proteasome regulators are depicted in orange. All proteins which were only enriched in the AX2/RFP-PSMD1/PSMA4-GFP-versus-NC comparison are depicted in dark yellow, in the ATG16^−^/RFP-PSMD1/PSMA4-GFP-versus-NC comparison in dark red, and in the ATG16^−^/RFP-PSMD1/PSMA4-GFP-versus-AX2/RFP-PSMD1/PSMA4-GFP comparison, in lilac. All other proteins are depicted in grey. Horizontal dashed line: *p*-value = 0.05, vertical dashed line: 3-fold enrichment.

**Figure 5 cells-12-01514-f005:**
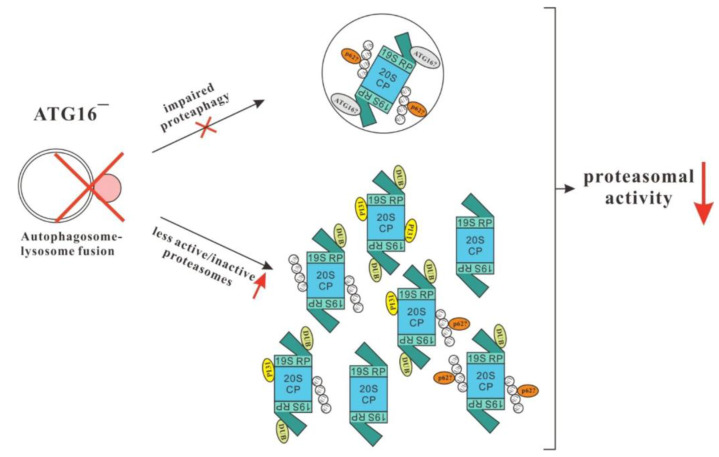
Model. In ATG16^−^ cells and many other autophagy-deficient strains, autophagosome-lysosome fusion is inefficient or strongly inhibited. As supernumerary, aged, or defective proteasomes are cleared by proteaphagy, the exchange of malfunctioning proteasomes is strongly inhibited. Proteasomes in these strains accumulate post-translational modifications, as, e.g., ubiquitination, which either per se may cause a decrease in activity or may constitute additional binding sites for proteasomal activity inhibiting proteins, as for example PI31. We propose that the relative increase of “aged”, less active or inactive proteasomes in autophagy-deficient strains results in the strong decrease in proteasomal activity and the observed fatal consequences for cellular homeostasis.

**Table 1 cells-12-01514-t001:** *D. discoideum* strains used in this study.

Strains	Summary	Reference
AX2	Axenically growing derivate of wild isolate NC-4	[43]
ATG9^−^	ATG9 knock-out mutant	[37]
ATG16^−^	ATG16 knock-out mutant	[27]
ATG9^−^/16^−^	ATG9 and ATG16 double knock-out mutant	[27]
AX2/RFP-PSMD1	AX2 cells expressing RFP-tagged PSMD1	[17]
AX2/RFP-PSMD1/PSMA4-GFP	AX2 cells expressing RFP-tagged PSMD1 and GFP-tagged PSMA4	This work
ATG16^−^/RFP-PSMD1	ATG16 null cells expressing RFP-tagged PSMD1	[17]
ATG16^−^/RFP-PSMD1/PSMA4-GFP	ATG16 null cells expressing RFP-tagged PSMD1 and GFP-tagged PSMA4	This work

**Table 2 cells-12-01514-t002:** Ubiquitinated proteasome-associated proteins detected by mass spectrometry in AX2 and ATG16^−^ cells.

	UniProt ID	DDB_G ID	Description	AA	Position	Mean Intensities (log2)	
						AX2	ATG16^−^
Exclusive in AX2	Q54SZ7	DDB_G0282111	Uncharacterized protein	K	700	24.51	0.00
	Q75JX9	DDB_G0272214	Uncharacterized protein	K	303	24.20	0.00
	Q54LV8	DDB_G0286389	60S ribosomal protein L34	K	106	21.81	0.00
Ubiquitinated in AX2 and ATG16^−^ cells	Q54VN6	DDB_G0280229	60S ribosomal protein L24	K	70	26.22	27.42
	Q55B20	DDB_G0271470	Uncharacterized protein	K	3	26.21	28.01
	Q86HT3	DDB_G0274439	BolA-like protein	K	19	25.16	27.28
	Q54KW5	DDB_G0287061	Uncharacterized protein	K	927	24.20	26.19
	**Q54QR2**	**DDB_G0283679**	**Proteasome subunit beta type-7**	**K**	**63**	**23.78**	**25.52**
	P14794	DDB_G0280755	Ubiquitin-60S ribosomal protein L40	K	48	23.61	25.40
Exclusive in ATG16^−^	B0G0Z1	DDB_G0270910	Eukaryotic translation initiation factor 4E	K	870	0.00	30.46
	Q54T94	DDB_G0281913	Uncharacterized protein	K	146	0.00	28.58
	Q55BM4	DDB_G0269190	T-complex protein 1 subunit alpha	K	468	0.00	26.14
	Q55GK4	DDB_G0267634	Uncharacterized protein	K	827	0.00	26.05
	Q54ED9	DDB_G0291538	Uncharacterized protein	K	3	0.00	25.71
	Q54HN5	DDB_G0289337	Uncharacterized protein	K	3	0.00	25.64
	Q55CG3	DDB_G0271028	Uncharacterized protein	K	18	0.00	25.50
	**Q54JB0**	**DDB_G0288187**	**DDI1 homolog**	**K**	**55**	**0.00**	**24.82**
	Q54QE4	DDB_G0283987	Bifunctional purine synth. protein purC/E	K	234	0.00	24.71
	Q8MMQ2	DDB_G0275539	Uncharacterized protein	K	86	0.00	24.64
	Q54GS9	DDB_G0289933	Uncharacterized protein	K	346	0.00	24.30
	Q54YL1	DDB_G0278621	Uncharacterized protein	K	604	0.00	23.89
	Q559T1	DDB_G0272438	Uncharacterized protein	K	231	0.00	23.73
	Q55C03	DDB_G0270284	Uncharacterized protein	K	373	0.00	23.67
	Q54WT5	DDB_G0279449	Uncharacterized protein	K	115	0.00	23.24
	Q54GX0	DDB_G0289871	Uncharacterized protein	K	2	0.00	22.15

Three independent experiments were performed with each strain and the precipitated proteins were subjected to mass spectrometry. Ubiquitinated proteasome-associated proteins were detected. The UniProt ID, the dictyBaseID (DDB_G), the modified amino acid (AA), its position in the protein and the log2 of the mean signal intensity are provided. K: lysine; AX2: Wild-type strain expressing RFP-PSMD1 and GFP-PSMA4; ATG16^−^: ATG16^−^ strain expressing RFP-PSMD1 and GFP-PSMA4. Proteins in bold, see Section 4.

## Data Availability

The mass spectrometry proteomics data have been deposited at the ProteomeXchange Consortium (http://www.proteomexchange.org/) via the PRIDE repository [33] with the dataset identifier PXD039713. All other data presented in this study are included in the article, and further inquiries can be directed to the corresponding authors.

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
