# Peer review of "Proteasomes of Autophagy-Deficient Cells Exhibit Alterations in Regulatory Proteins and a Marked Reduction in Activity"

_cells, 2023, doi:10.3390/cells12111514_

Round 1
Reviewer 1 Report (Previous Reviewer 3)
The changes made to the manuscript by the authors to address other reviewer’s concerns are appreciated. In regards to the my previous concerns that the analysis is primarily descriptive the authors have also argued the significance of their previous and current results. Unfortunately protein abundance changes in mutant cells does not verify the importance of those proteins in the mutant phenotype. This reviewer still believes the conclusions about the importance of these proteins in the described phenotype is still speculative.
Reviewer 2 Report (Previous Reviewer 1)
My comments have been addressed or satisfactory explanations given.
This manuscript is a resubmission of an earlier submission. The following is a list of the peer review reports and author responses from that submission.
Round 1
Reviewer 2 Report
Xiong et. al. explore an intriguing inconsistency between proteasome abundance and activity when comparing autophagy defective Dictyostelium cells with wild type. Previous work from this group showed that while proteasome abundance, as measured by two proteasome subunits, remained consistent when comparing wild type Dictyostelium discoideum to Atg9 or Atg16 mutants, proteasome activity was significantly reduced. Here, using pull down assays and mass spectrometry, the authors conclude that in fact all proteasome subunits remain abundant in an Atg16 mutant strain. The main question being addressed is why proteasome activity is reduced in these mutants. The data presented, however, do not answer this question. Instead, the data are descriptive of what proteins are enriched following a proteasome immune precipitation using GFP-Trap in the autophagy mutant vs the wild type. While the authors speculate about the potential mechanisms based on the enriched proteins they find, for example, the depletion of ubiquilin shuttle factor or the enrichment of the inhibitor PI31, there is no experimental data revealing a mechanism of a reduction in proteasome autophagy. This, combined with concerns regarding the immunoprecipitation conditions used, make this work premature for publication in my opinion.
Measuring the activity of the purified proteasomes using a plate reader assay similar to what this group has done in the past would strengthen their conclusions and show that what they purified behaves same as lysates they measured before. This would help to confirm that proteasomes themselves are less active (as the authors propose for “aged” proteasomes).
Figure 2 besides tagged PSMA4 a wild type copy is still present based on western blotting? In the materials and methods, and when first discussed in the paper, it should be made clear that the PSMA4-GFP construct being expressed is overexpression of the protein. This could affect the interpretation of the results. For example, a mixture of proteasomes would result in vivo, with some 20S particles having no, one, or two copies of PSMA4-GFP. There is also more than double the WT amount of PSMA4 as can be seen in Fig. 2A.
Figure 3. IP shows mainly CP subunits and no substantial amount of RP subunits that are detectable with silver stain. The IP lysis buffer does not appear to contain ATP. The RP-CP association generally is not stable without nucleotide present for the ATPases to bind. While the mass spectrometry might still detect RP subunits and compared to the negative control RP this might still be enriched, most RP has clearly been lost during purification (unless > 90% of the proteasomes in Dicty would be 20S). This is a major concern for any quantitative comparison and conclusions regarding non-CP subunits that follows (like in figure 4).
“The data provided no explanation for the observed reduced proteasomal activity, as there was no indication for a reduced number of assembled proteasomes in the ATG16 ̄strain.” This conclusion based on the IP-mas spec is hard to determine, if there is an excess of input material, differences in number of proteasomes might be lost. Not clear that a careful quantitative IP was conducted to justify this conclusion.
Line 275 “We conclude that the strong decrease in proteasomal activity in the ATG16 ̄strain is not due to a decrease in the number of proteasomes in this strain. Based on our results, the ATG16 ̄strain harbors at least similar amounts of fully assembled 26S proteasomes (Figure 4; Table S1). “ I don’t agree with this conclusion. Based on western I agree levels of proteasome subunits are similar. Based on IP with mass spec, composition of CP does not appear different. Amounts of 20S versus 26S in cell remains to be determined and composition of RP cannot be properly assessed from the experiments as explained above.
Line 378: “…could result in reduced feeding of the proteasome and thus be partially responsible for the reduced proteasomal activity in this strain.” I don’t understand this argument. The measured proteasome activity was for small peptide substrates if I understand correctly. These do not rely on being fed to the proteasome.. so ubiquilin levels should not matter for that activity.
Line 421 “Therefore, we also considered differences in ubiquitination as a possibility for inhibition of proteasomal activity in ATG16 ̄cells.” I would agree if the assay used for proteasome activity was based on ubiquitinated substrates and not fluorogenic peptides…. this however does not seem to be the case looking at prior publications measuring the activity. Maybe, it could be a contributing factor to accumulation of ubiquitinated proteins, but it is hard to dissect such accumulation from accumulation of ubiquitinated material due to lack of autophagy….
Line 287 enrichment of PI31, which is an inhibitor of the proteasome, as explanation of inhibition. Also, Line 386 “We noticed a more than 386 16-fold enrichment of PI31 with proteasomes of ATG16 ̄versus AX2 cells, which could be 387 a major reason for the reduced proteasomal activity in ATG16 ̄ cells.” This is an interesting increase, and the interaction is not dependent on ATP. However, from the data it is unclear what the stoichiometry for these proteins is. PI31 inhibits by direct binding, so, to explain a 90% reduction in activity, this protein should be highly enriched and almost stoichiometric with CP subunits. It would be valuable to know the stoichiometry of PI31 vs 20S to really understand how this contributes to the reduced activity. For example, if it goes from binding 0.5% of proteasomes to 8% that still would limit the ability to explain the difference in activity.
Minor:
Line 16 For clarity with two “and” following, rephrase to something like: for the clearance and recycling of proteins and organelles in eukaryotic cells
Line 210 “extremely significant” significant is statistical term… so what is extremely significant… better state the number of use different term dramatic reduction in activity.
No legends for supplemental figures.
For consistency in figures, all gels and westerns should be boxed or not.
Some westerns (ex. Fig. 3B) should be contrast adjusted
Line 282 “enriched in the ATG16 ̄versus AX2 comparison” isn’t that redundant with the statement seven “uniquely enriched proteins “ stated in line 281.
Reviewer 3 Report
This study examines differences between wild-type and autophagy mutants. Previous analysis of cells with disrupted genes encoding proteins involved with autophagy (ATG9 or ATG16) resulted in decreased proteasomal activity and an increase in poly-ubiquitinated proteins but no significant changes in proteasomal subunits. The authors expressed and immunoprecipitated a PSMA4-GFP protein in these cells and analyze associated proteins using mass spectrometry. They notice differences in some proteasomal regulators and differential ubiquitination and now speculate that these differences might account for the decrease in proteasomal activity. The speculation might be correct but at this point remains a correlation rather than a cause and effect. Further analysis, such as mutating or overexpressing these proteasomal regulators, would be important to test their roles in autophagy mutants. The data in the manuscript is descriptive but does not provide a strong basis for a mechanistic explanation for the initial question asked by this study.